# Literacy and Mental Health of Portuguese Higher Education Students and Their Use of Health Promotion Strategies during Confinement in the COVID-19 Pandemic

**DOI:** 10.3390/ijerph192114393

**Published:** 2022-11-03

**Authors:** Ana Paula Oliveira, Joana Rita Nobre, Henrique Luis, Luis Soares Luis, Núria Albacar-Riobóo, Lara Guedes Pinho, Carlos Sequeira

**Affiliations:** 1Health School, Polytechnic Institute of Portalegre, 7300-555 Portalegre, Portugal; 2Faculty of Nursing, University of Rovira e Virgili, 43003 Tarragona, Spain; 3Unidade de Investigação em Ciências Orais e Biomédicas (UICOB), RHODes-Rede de Higienistas Orais para o Desenvolvimento da Ciência Faculdade de Medicina Dentária, Universidade de Lisboa, Rua Teresa Ambrósio, 1600-277 Lisbon, Portugal; 4Center for Innovative Care and Health Technology (ciTechcare), Polytechnic of Leiria, 2410-541 Leiria, Portugal; 5School of Health Sciences, Polytechnic of Leiria, 2410-541 Leiria, Portugal; 6Nursing School, University of Evora, 7000-811 Evora, Portugal; 7Comprehensive Health Research Center, 7002-554 Evora, Portugal; 8Nursing School of Porto, 4200-072 Porto, Portugal; 9Group Inovation and Development in Nursing (NursID), Centro de Investigação em Tecnologias e Serviços de Saúde (CINTESIS), 4200-450 Porto, Portugal

**Keywords:** mental health, literacy, higher education students, health promotion strategies, confinement

## Abstract

The pandemic of COVID-19 caused significant changes in economies and societies with a major impact on the entire education process. However, these changes did not invalidate a constant effort of adaptation. This cross-sectional, descriptive, and correlational study used an online questionnaire administered to students from various study cycles at higher education institutions in Portugal, with the aim of exploring the influence of literacy and mental health on the use of mental health promotion strategies during COVID-19 confinement. A total of 329 students from higher education institutions participated in this study, mostly from the age group 18–24 years (*n* = 272; 82.7%) and female (*n* = 265, 80.5%). The most mentioned health promotion strategies during this period included studying (*n* = 170; 51.7%); physical activities (*n* = 151, 45.9%); social networking (*n* = 124, 37.7%); cooking activities (*n* = 120, 36.5%); and listening to music (*n* = 118, 35.9%). Academic success is self-reported, and it is weakly correlated with the MHI5 (r = 0.103, *p* = 0.063). Students in the pre-graduate programs studied more during the times of the pandemic and used this activity as a mental-health-promoting strategy with a statistically significant difference (*p* = 0.033). Although it was difficult to improve health literacy related to COVID-19 in such a short period of time, there was a very strong motivation to access, understand, evaluate, communicate, synthesize, and apply information and knowledge to maintain mental health through self-care using health promotion strategies.

## 1. Introduction

The impact of SARS-CoV-2 on the world’s society has been profound. Since late December 2019, a novel coronavirus with human-to-human transmission emerged in China [1]. The new disease named COVID-19 had a tremendous impact leading to a global lockdown with dramatic changes in daily life. As of 16 October 2022, 621 million confirmed cases and 6.5 million deaths have been reported globally [2].

As a result of the lockdown, millions of students were not able to continue their daily activities at school with colleagues and friends, since early on the impact of COVID-19 on literacy was relevant. This impact is relevant to study and to look for the relationship among the variables of literacy and mental health related to health promotion strategies, since literacy levels can change according to study cycles, and the relationship is recognized between higher levels of literacy and higher levels of creating mental health promotion strategies. This research is relevant to show if this happens in this specific population of alto Alentejo students. This may contribute to identifying theoretical and educational strategies both for students and teachers in order to empower the higher education community for better mental health.

The term literacy is increasingly used in a broader sense than its original meaning (the ability to read and write). According to the United Nations Educational, Scientific and Cultural Organization [3], the term “literate” is mainly meant to be “familiar with literature” in general “well-educated, educated”, and there are more than 250 different definitions of literacy [4]. Functional literacy comprises a set of technical and social skills necessary for the organization of an individual’s life in society [5]. General literacy will give people fundamental skills such as decision making and civic and personal responsibility [6]. In the late 19th century, literacy began to refer not only to the ability to read and write but also to an individual’s skills in the complex issues of health promotion and maintenance in modern society [7].

Making a connection between general literacy, the set of individual skills, and individual health is called health literacy. Health literacy can be defined as “the degree to which individuals can obtain, process, understand and communicate about health-related information needed to make informed health decisions” [8]. Some of these skills include numeracy, reading, writing, being able to express one’s own ideas, communicate effectively, and use technologies [5]. The concept of health literacy began in the 1970s and is very flexible, allowing anyone to identify almost anything they want as health literacy. The conceptualization of health literacy emphasizes the relevance of going beyond the personal dimension and considering the interaction between the demands of health systems and the individual’s abilities [9]. Liu, C. et al. in 2020, referring to the concept of health literacy, extracted three key themes representative of the various concepts adopted in the included studies: (1) health knowledge, health, and health systems; (2) processing and using information in various formats in relation to health; and (3) ability to maintain health through self-management and working in partnership with health professionals. They also mentioned that the synthesis of information was not included, and this dimension is particularly important because nowadays people are flooded with a huge amount of information and must have the ability to compare, synthesize and evaluate in order to integrate the various information to make informed and correct decisions [10]. It is recognized that a higher level of education and a correspondingly higher level of literacy can be a predictor of higher levels of health literacy. A higher level of literacy allows an individual to use information about health issues as a tool to improve their own health, also skills such as reading and writing are essential for a good relationship with health education and self-management of health, as well as decision making [11]. So, students in higher education and in higher academic years may have a greater ability to promote their mental health because of the higher level of health literacy they can attain. However, it should be considered that it is necessary to work with students not only on increasing their level of health literacy but also on the wide range of knowledge fields that contribute to health literacy [12]. These authors also mention that the type of access to higher education has an influence on the level of health literacy at the entry to this degree, which may lead to different health-promoting approaches throughout the training.

Thus, it is critical to consider the skills that already exist, and their use for health promotion, as the association between health literacy and mental health is clear. According to the World Health Organization (WHO), mental health is “a state of well-being in which the individual realizes his or her own abilities, can cope with the normal stresses of life, can work productively and fruitfully, and is able to make a contribution to his or her community” [13]. As with health literacy, individual empowerment is critical to increasing mental health literacy, which can be defined as the “knowledge and beliefs about mental disorders that aid their recognition, management, or prevention” [14]. As such, mental health literacy has been conceptualized as comprising four distinct but related components: (1) understanding how to obtain and maintain good mental health; (2) understanding mental disorders and their treatments; (3) decreasing stigma related to mental disorders; (4) improving help-seeking effectiveness (knowing when, where, and how to obtain good mental health care and developing the necessary skills for self-care) [15].

Social media also plays an important role in young people’s literacy, even when it comes to representations of health aspects and images [16]. Today, social media and information technology are key to helping adolescents and young adults acquire the skills critical to promoting their mental health, and students in higher education are one of the groups receiving increasing attention from experts [17]. In a study conducted in Chile, a strong, inverse, and statistically significant association was identified between education and mental disorders, and students with a high degree of anxiety had less satisfactory academic results and when this anxiety was reduced, there was an improvement in their performance [18].

The pandemic of COVID-19 caused significant changes in economies and societies with a major impact on the entire education process. Most schools closed, and it is estimated that at the peak of the crisis, 94% of the world’s student population was out of school and at home. Information was accessed through the Internet from all sources, but not all people were able to make the best use of this information in their self-care and health promotion strategies. Some authors argue that people with better health literacy can more effectively distinguish the accuracy and authenticity of information [19,20,21], so literacy can be associated with health indicators [22].

Therefore, this pandemic had, and still has, a profound impact on the literacy of young people. A study conducted in the United States to assess the effects of the COVID-19 pandemic on the mental health of college students found that most of these students had increased stress and anxiety due to the pandemic, using support from others and adopting coping mechanisms to deal with stress and anxiety, yet more than half reported that they were unable to cope adequately [23]. These two symptoms are the most frequently reported by university students when asked about the effect of the pandemic on their mental health [24].

A study in Switzerland found that college students were on average more depressed, slightly more anxious, more stressed, and lonelier than before the pandemic. However, there were also positive aspects, such as reduced fear of failure and competition among students [25]. Similar results were found in a study in Greece, where an increase in the quantity and decrease in the quality of sleep was reported, and students generally felt that their quality of life worsened [26], especially in health-promoting behaviors [27].

The pandemic caused by COVID-19 has concerned, and continues to concern, health authorities around the world, for what it has already caused, and for the consequences associated with protective measures, especially quarantine: reduced freedom, reduction in economic activity in some sectors and in usual routines or livelihoods, and has led to increased levels of loneliness, depression, harmful use of alcohol and drugs, and self-injury or suicidal behavior. Preventive behaviors have helped to reduce the spread of COVID-19, but conventional individual health behaviors, such as exercise and diet, have also played an important role in maintaining physical and mental health during the COVID-19 pandemic [28].

Students in higher education also experienced changes in their academic and personal experiences, being limited in direct contact with their colleagues and professors, face-to-face classes were replaced by distance learning, restricting contact with other people and direct access to academic services, affecting their physical and mental well-being, and some experienced considerable psychological problems [29], such as stress, anxiety, depressive thoughts, fear, and worry about their own and their family members’ health [23].

However, these changes did not invalidate a constant effort of adaptation so that the difficulties experienced could be overcome using several self-care and health promotion strategies.

Considering the above and the relationship between literacy, mental health, and health promotion strategies, we have identified the following research questions:(a)What is the association between mental health and health promotion strategies of higher education students in the Alentejo region of Portugal during confinement during the COVID-19 pandemic?(b)What is the correlation between self-reported academic success and mental health of higher education students in the Alentejo region of Portugal during confinement during the COVID-19 pandemic?(c)What is the association between the cycle of studies and mental health promotion strategy in higher education students in the Alentejo region of Portugal during confinement during the COVID-19 pandemic?

In this context, the aim of this study is to explore the influence of mental health literacy on the use of mental health promotion strategies during COVID-19 confinement.

## 2. Materials and Methods

This study was developed during the COVID-19 confinement period with the research question to identify the most commonly used mental health promotion strategies and their relationship to mental health in adults using the Mental Health Inventory-5 (MHI-5), which is a brief, valid, and reliable instrument that also includes characterization questions.

### 2.1. Study Model

This cross-sectional, descriptive, and correlational study used an online questionnaire administered to students from various study cycles at higher education institutions in Portugal.

The study hypothesis was to evaluate if there was any relationship between the literacy and mental health of Portuguese higher education students and their use of health promotion strategies during confinement during the COVID-19 pandemic. Independent variables were defined by literacy (ability to read, write, speak, and compute and solve problems at a level of proficiency that allows function in society and the development of one’s own knowledge and potential), health literacy (“the degree to which individuals can obtain, process, understand and communicate about health-related information needed to make informed health decisions” [8]), and mental health (“a state of well-being in which the individual realizes his or her own abilities, can cope with the normal stresses of life, can work productively and fruitfully, and is able to make a contribution to his or her community” [13]). The dependent variable is the use of health promotion strategies (processes to enable and increase one’s control over health in order to improve it) during the COVID-19 pandemic. Other independent variables are measured by the participation of the students in several activities or groups of social networks (networks that facilitate social contact, leisure, and pleasure), in a huge variety of applications and activities, and academic achievement as the extent to which a student achieves educational goals.

### 2.2. Data Collection

A non-probability convenience sampling was obtained, calculated for a 5% margin of error and a 90% confidence level, requiring 256 participants. The questionnaire was created using Google Forms and administered online via email with the access link for completion and sent to the 4450 students with active enrollment in the institutions, and 329 valid questionnaires were obtained.

In the questionnaire, the study was described, and the participant could only continue to participate after giving their consent to participate. Ethical issues were always safeguarded. Data confidentiality was guaranteed, and the data collected were stored in the researchers’ personal folders with security codes. The study was approved by the Ethics Committee of Polytechnic Institute of Portalegre (Ethics Opinion n°. SC/2020/316 of 20/02/2020) and by the Data Protection Officers of both institutions.

Data collection took place between 15 April and 20 May 2020.

The questionnaire included a sociodemographic characterization to characterize the sample. Mental health was also studied using the Mental Health Inventory in a reduced version (MHI-5), translated and validated for Portugal by Pais Ribeiro in 2011 [30]. The reduced version of the Mental Health Inventory (MHI-5), translated and validated for Portugal was used to assess mental health. Based on the thirty-eight-question inventory, a reduced version was developed called Mental Health Inventory 5 (MHI-5), composed of five items representing four dimensions of mental health: Anxiety, Depression, Loss of Emotional–Behavioral Control, and Psychological Well-Being (Ware et al., 1992 [31]). The scale’s rating is obtained by summing the items (two items with the rating reversed). Higher levels in the summation correspond to better mental health between 5 and 30.

The Mental Health Inventory was initially developed within the Rand Health Insurance Experiment (HIE), a 15-year study initiated in 1971 for the United States Department of Health, Education, and Welfare that evidenced the existence of a positive dimension (psychological well-being, positive mental health status) and a negative dimension (psychological distress, negative mental health status) [30].

Veit and Ware, in 1983, developed the Mental Health Inventory (MHI) to assess psychological stress and well-being in the general population and not only in people with mental illness [30]. Based on this thirty-eight-question inventory, a shortened version called Mental Health Inventory 5 (MHI-5) was developed, which consists of five items representing four dimensions of mental health, namely Anxiety, Depression, Loss of Emotional–Behavioral Control, and Psychological Well-Being [31]. These five items have, in the original study, a correlation of r = 0.95 and r = 0.92, with the total score of the 38-item version. The Portuguese adaptation shows a correlation of r = 0.95 between the MHI-5 and the MHI-38. The scale score is obtained through the sum of the items (2 items with inverted scores). Higher levels in the sum correspond to better mental health between 5 and 30. Numerous investigations have shown that the MHI-5 is a useful screening test in the assessment of mental health [30].

### 2.3. Statistical Analysis

Descriptive statistics (absolute and relative frequency, mean, and standard deviation) were used according to the type of variable to characterize the sample under study.

To measure the association between two quantitative variables, a correlation was used; in the cases of interval or ratio variables and distributions approaching normality, Pearson’s coefficient was used; and in ordinal variables or when distributions departed from normality, Spearman’s coefficient was used. To evaluate the strength of the association, intensity levels were used: low correlation (0.21 to 0.39); moderate (0.41 to 0.69); high (0.71 to 0.89); very high >0.90 according to Marôco (2011) [32]. The chi-square test was used to compare the proportions between the study variables and the demographic characteristics analyzed. Data analysis was performed using the computer program SPSS version 27 with a significance level of 5%.

## 3. Results

A total of 329 students from higher education institutions participated in this study, mostly from the age group 18–24 years (*n* = 272; 82.7%). Most of the participants were female (*n* = 265, 80.5%). The distribution by gender and age group is shown in Table 1.

Students were asked to indicate a self-report of their academic success.

Academic success is self-reported from mediocre to very good (*n* = 4, 1.2% mediocre; *n* = 59, 18% sufficient; *n* = 220, 66.9% good; and *n* = 46, 13.9% very good). It is weakly correlated with mental health (r = 0.103, *p* = 0.063), meaning that students who report better grades are those who have better mental health, although this is a weak non-significant correlation.

Most students (*n* = 274, 83.2%) attend undergraduate courses and in any of the degrees attended (professional higher technical course, bachelor, master or postgraduate) there is no statistically significant difference with mental health (*p* = 0.384).

The most mentioned health promotion strategies during this period included studying (*n* = 170; 51.7%); physical activities (*n* = 151, 45.9%); social networking (*n* = 124, 37.7%), cooking activities (*n* = 120, 36.5%); and listening to music (*n* = 118, 35.9%).

There is also a low negative correlation (ρ = −0.036), which is statistically non-significant (*p* = 0.512), between mental health assessed by the MHI-5 and participation in academic groups. Regarding the relationship with participation in recreational groups, there is a low positive correlation (ρ = 0.016) that is statistically non-significant (*p* = 0.777). The same is true for participation in religious groups (ρ = −0.076; *p* = 0.171).

Considering the relationship between mental health, measured by the MHI-5, and the type of social networks used during confinement, it is found that the correlations are low and not statistically significant, as described in Table 2.

The correlation between the MHI5 and the time of use of social networks in confinement showed a low negative correlation (ρ = −0.174), but it was statistically significant (*p* = 0.001), i.e., participants with higher MHI5 had a shorter time of use of social networks.

Furthermore, in the study of the relationship between the MHI5 and the reasons for using social networks, it was observed that only the “meet other people or make new friends” with a low negative correlation (ρ = −0.119) was statistically significant (*p* = 0.031), Table 3.

Students in the pre-graduate programs studied more during the times of the pandemic and used this activity as a mental-health-promoting strategy with a statistically significant difference (*p* = 0.033).

Considering the relationship between the frequency of course type (pre- and postgraduate) and the type of social networks used during confinement, correlations are found to be low and only statistically significant for the social network Instagram as described in Table 4.

In the study of the correlation between the type of course attended (pre- and post-graduation) and the time spent using social networks in confinement, no statistically significant differences were found for the time spent on the different social networks.

In the analysis of the relationship between the level of education attended (pre- or post-graduation) and the reasons for using social networks, a low negative correlation (ρ = −0.123) that was statistically significant (*p* = 0.026) was found for the activity “Read curiosities, news, get information”, Table 5.

## 4. Discussion

The purpose of the present study is to explore the influence of literacy and mental health on the use of mental health promotion strategies during COVID-19 confinement and found that there is a weak correlation between academic achievement and mental health, which agrees with the results found by King et al. who state that students with lower mental health are the worst academic performers [33]. In 2018, Lipson also reports that mental health problems were a predictor of academic failure [34]. Another study also conducted during the pandemic period revealed that young people had higher levels of anxiety and depression and that men and women had similar levels of anxiety and depression during the year, with women having higher levels of anxiety [35]. We also found that there was no statistically significant difference between the academic degree attended and mental health, which goes against the findings by Duffy et al. who state that there are clinically significant mental health symptoms among undergraduate students impacting academic performance [36], which is corroborated by Bennett et al. who state that male postgraduate students have lower mental health problems, particularly about depression [37].

It was observed in the present study that the main health promotion strategies reported during the pandemic period were studying (51.7%); physical activity (45.9%); social networking (37.7%); cooking (36.5%); and listening to music (35.9%). The study by Taeymans et al. found similar evidence for physical activity practice, namely for postgraduate students [38]. However, not all studies suggest this evidence as is the case of that reported by Luciano et al. according to which a decrease in physical activity and an increase in sedentary lifestyles were observed in undergraduate students [39]. Regarding other health promotion strategies cooking activities and studies were also mentioned by Cruyt et al. as being among the most developed during the pandemic despite not being statistically different from what was conducted in the pre-COVID period [40]. In this study, no statistically significant differences were ever observed for participation in recreational, academic, or religious groups and mental health; however, this is not reported by Kim who mentions that individuals who participated in group activities with cultural, artistic, and social characteristics had protective features of mental health [41].

As we found, Berryman, in 2018, also found no relationship between social media use and mental health [42]. Regarding literacy, the literature shows that a higher level of literacy and therefore health literacy moderates the relationship between social networks and health-promoting behaviors. Greater use of social networks is also associated with greater health promotion [43]. It is also found that among social networks, user-oriented social networks, such as Facebook, are preferred over more content-oriented or professional social networks, for example, YouTube and LinkedIn. This is predictive of the type of information that individuals will obtain, and may, according to Hu, influence behaviors in the context of the COVID-19 pandemic [44]. The present study observed that participants with a higher level of MHI-5 had a shorter time of social media use; however, it is suggested that the time of social media use is not as important in terms of mental health as the frequency of social media use, namely with regard to preventive behaviors for COVID-19 [43]. This author also found that the level of literacy, namely health literacy and knowledge of the disease (COVID-19), positively influenced the relationship between social media use and health-promoting behaviors.

The observed data indicate that undergraduate students studied more during their confinement time and used this activity as a mental health strategy. This pattern of increased studying was also verified by Hendriksen during confinement [45]. A statistically significant relationship was observed in the present study indicating that undergraduate students looked to social media to read, look for curiosities, watch the news, and get information. Ranjbar et al. reported that the main preferences of students during the confinement period were computer games, studying, and watching television, which may be similar to our study only in watching the news and getting information through the media, namely television [46].

## 5. Conclusions

The pandemic times were a challenge for everyone, and higher education students coped with it by using health promotion strategies mainly related to school, physical well-being, and social interaction. The rapid emergence of the pandemic and the confinement we were all forced into, led to new and great learning through any means possible to minimize the negative impact of this novelty. Everyone wanted to know more. Although it was difficult to improve health literacy related to COVID-19 in such a short period of time, there was a very strong motivation to access, understand, evaluate, communicate, synthesize, and apply information and knowledge to maintain mental health through self-care using health promotion strategies.

Our study suggests that Portuguese higher education students with higher health literacy may more actively adopt health promotion strategies and self-care, and these strategies may be enhanced with awareness-raising actions to improve health behaviors; however, it is relevant to notice that this is a study performed only on Portuguese students with the country specificities. We observed that within this scope of health promotion, school-related activities were performed, as well as physical activities, the use of social networks, cooking activities, and other recreational activities such as music. We should point out that a statistically significant correlation was found between less time spent using social networks and students who had better levels of mental health. It is also worth noting that, also statistically significantly, students with a higher MHI5 level did not use social networks to expand their network of friends during the confinement. We also found that there was no statistically significant difference between the academic degree attended and the mental health level who did not use social media to expand their friendship network during confinement. The undergraduates were the ones who used the study activity in a statistically significant way as a mental-health-promoting activity, compared to the postgraduates. The development of health literacy is crucial to the promotion of mental health because better-informed and better-prepared students will surely be better able to face the adversities they will encounter. The universities must include mental health policies in their strategic plans. Specific diagnostic and intervention programs should be defined, as well as programs, to promote the mental health of university students. These intervention programs should be transversal to all students, with particular attention to the most psychologically vulnerable students, female, younger, out-of-home, and with scholarships (usually the most financially needy) [47]. With this study, we hope to contribute to the development of educational strategies, encouraging other researchers to further expand their knowledge in this area of health strategies for mental health promotion.

Besides being applied only to Portuguese students, another limitation of our study is the fact that it is a cross-sectional study, not allowing a cause–effect relationship. We cannot conclude from this study whether it is literacy and mental health that lead to the adoption of health promotion strategies; however, it shows the relevance to promote the mental health of higher education students to protect them and encourage a healthy lifestyle. Another limitation could be the impossibility to transpose these results to a situation other than a confinement for health reasons. Thus, the evidence seems consistent that both higher years of study and field of study are related to mental health literacy; however, these studies are not common, and additional variables directly related to college experience should be investigated for a more comprehensive understanding of the factors associated with mental health literacy in the college population.

## Figures and Tables

**Table 1 ijerph-19-14393-t001:** Distribution by gender and age group.

		Age					
		18–24	25–30	31–35	36–44	≥44	Total
		Years	Years	Years	Years	Years	
Gender	Male	48	8	3	4	1	64
	Female	224	21	5	10	5	265
	Total	272	29	8	14	6	329

**Table 2 ijerph-19-14393-t002:** Correlation values and significance between MHI5 level and social network typology.

	Facebook	WhatsApp	Instagram	Twitter	LinkedIn
Correlation	0.046	0.058	−0.097	−0.102	0.060
Significance	0.406	0.291	0.079	0.075	0.276

**Table 3 ijerph-19-14393-t003:** Correlation values and significance between mental health level and reasons for using social networks.

	Contact Family and Friends	Work	Meet Other People or Make New Friends	Play	Sharing Life with Others (Travel, Photos, Meals)	Get to Know Other People’s Lives (Travel, Photos, Meals)	Read Curiosities, News, Gather Information
Correlation coefficient	0.016	−0.044	−0.119 *	−0.074	−0.063	−0.046	0.080
Signifiance	0.778	0.427	0.031	0.182	0.257	0.408	0.147

* *p* < 0.05

**Table 4 ijerph-19-14393-t004:** Correlation values and significance between course type and social network typology.

	Facebook	WhatsApp	Instagram	Twitter	Linkedin
Correlation coefficient	−0.029	0.034	−0.265 **	−0.067	0.046
Significance	0.601	0.541	<0.001	0.249	0.404

** *p* < 0.001

**Table 5 ijerph-19-14393-t005:** Correlational and significance values between the type of course and reasons for using social networks.

	Contact with Family or Friends	Work	Meet Other People or Make New Friends	Playing	Sharing Life with Others (Trips, Photos, Meals)	Get to Know Other People’s Lives (Trips, Photos, Meals)	Read Curiosities, News, Get Information
Correlation coefficient	−0.066	0.065	−0.005	−0.055	0.069	0.050	−0.123 *
Significance	0.230	0.237	0.924	0.323	0.212	0.362	0.026

* *p* < 0.05

## Data Availability

Data available on request due to ethical restrictions.

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
