# Peer review of "Literacy and Mental Health of Portuguese Higher Education Students and Their Use of Health Promotion Strategies during Confinement in the COVID-19 Pandemic"

_ijerph, 2022, doi:10.3390/ijerph192114393_

Round 1
Reviewer 1 Report
1. The title seems to be inconsistent with the content?
2. In the Introduction section: The author had organized a lot of literature, but the content of the literature is not systematic. I can’t understand the relationship between the research variables and the importance of this research.
3. Several issues of research method are as follows:
1)This article lacks a clear research model, research questions and research hypotheses.
2)The definition of independent and dependent variables was not clear. The author did not clearly define research variables such as "literacy", "health promotion strategies", "social network typology", "reasons for using social networks", "academic achievement", etc. The author did not explain how these variables are scored. Is it appropriate for students to self-assess their academic achievements? In addition, are the scores of these variables suitable for correlation analysis with mental health?
3)The author did not specify the population and the specific sampling method.
4. The reliability and validity of research tools are very important. It is common to translate scale into a new language and use it in different countries. The author can explain the process of translating the original scale into Portuguese and how to ensure that items retain their original meaning after translation. It is common practice to use CFA (Confirmatory Factor Analysis) to further illustrate the validity of the scale.
5. It is suggested that the authors could interpret the findings more and generalize them into "theoretical implications" and "educational implications," especially for higher education supervisors or teachers and higher education students. In this way, the value and contribution of this research can be further demonstrated.
6.It is suggested that authors may state "Limitations and directions for future research".
Author Response
The authors appreciate and acknowledge the reviewers work and very meaningful comments. We respectfully answer to those questions:
Taking in consideration the comments from reviewer 1, the title comment was accepted, it was changed and now reads “Literacy and mental health of Portuguese higher education students and their use on health promotion strategies during confinement in the Covid19 pandemic” making it more consistent with the article content. Regarding the introduction changes were made in order to organize the introduction text in a systematic way making clear the relationship between the variables and the importance of the study. The research question was included in the methods section. Study model is now in point 2.1 of the methods as well as the research hypothesis. Independent and dependent variables were made clear in the methods section. The population and the sampling id described under the sub-chapter Data collection. The instruments to collect data were not translated by the present article authors, the translation and validation process is described in the literature that is presented as references in this article. Implications for higher education institutions and students were presented in the conclusions. Limitations were presented in conclusions
Reviewer 2 Report
General
You need to make clear what is the difference between terms mental health, health literacy and mental health literacy.
1. Introduction
The first two sentences in your Abstract section gave me the impression that you are going to structure this part differently. Namely, firstly focusing on COVID 19 pandemic in relation to this subject and then on defining the terms.
I suggest that you restructure this part in sub-chapter to make so that the readers can understand your thoughts clearly (a short paragraph with a few introductory words like COVID 19 part that you have in your abstract, and then you could continue with, e.g., definition of literacy and healthy literacy, mental health promotion strategies….). You should take into account that each subchapter should have at least two paragraphs, therefore, you do not need to divide this part into many subchapter (they need to provide information about one issue)
2. Methods
You should take into account that each subchapter should have at least two paragraphs.
You need to specify what variable you have included in your research before you start describing them.
3. Results
You should recheck your reference to table 2.
Your results are confusing. If possible, you should provide the data mentioned as a part of the text in tables.
You should present data in your tables 2 to 5 clearly. Your results show only correlations. Is it possible to do a more complex data analysis then simply calculating correlations?
4. Discussion
Before starting with summary of your findings you should repeat the goal of your paper (just to remind readers about it).
Each paragraph should consist of two or more sentences.
5. Conclusions
I prefer that authors state clearly the scientific contribution of the paper, practical implication, limitations of the research and recommendations for future research.
Author Response
The authors appreciate and acknowledge the reviewers work and very meaningful comments. We respectfully answer to those questions:
Reviewer 2 presented us relevant comment that we appreciate, we made clear the difference between terms like mental health, health literacy and mental health literacy. The introduction was restructured according to the requested by the reviewer. The authors have considered the comment on the need to have at least two paragraphs in each section of methods. In the method section the variables are now specified. The tables were not correct, and it was done in order to meet what was described in the text. The goal was added at the beginning of the discussion. Implications for higher education institutions and students were presented in the conclusions. Limitations were presented in conclusions
Reviewer 3 Report
In the article presented for review, the set goal, which is: „to explore the influence of literacy and mental 140 health on the use of mental health promotion strategies during Covid 19 confinement” was achieved. Also, the results of the research were clearly presented. They come from the adopted research tool [Mental Health Inventory-5 (MHI-5)]. Conclusions result from the conducted research. However, the results of the research may only be of local significance, depending on the specificity of the country. Therefore, the limitations of the research process and inference should be clearly indicated.
Author Response
The authors appreciate and acknowledge the reviewers work and very meaningful comments. We respectfully answer to those questions:
Considering Review 3 suggestions the specificity of the study is now clearly indicated in the title, mentioning that it is about Portuguese students. And also, the specificity of Portugal was added as a limitation of the study.
Round 2
Reviewer 1 Report
The author improved comment 1 and did not sufficiently revise comments 2-6.
I recommend that the author indicate the line number of the revised text when answering each comment.
I hope the author's efforts will make this article better.
2. In the Introduction section: The author had organized a lot of literature, but the content of the literature is not systematic. I can’t understand the relationship between the research variables and the importance of this research.
3. Several issues of research method are as follows:
1)This article lacks a clear research model, research questions and research hypotheses.
2)The definition of independent and dependent variables was not clear. The author did not clearly define research variables such as "literacy", "health promotion strategies", "social network typology", "reasons for using social networks", "academic achievement", etc. The author did not explain how these variables are scored. Is it appropriate for students to self-assess their academic achievements? In addition, are the scores of these variables suitable for correlation analysis with mental health?
3)The author did not specify the population and the specific sampling method.
4. The reliability and validity of research tools are very important. It is common to translate scale into a new language and use it in different countries. The author can explain the process of translating the original scale into Portuguese and how to ensure that items retain their original meaning after translation. It is common practice to use CFA (Confirmatory Factor Analysis) to further illustrate the validity of the scale.
5. It is suggested that the authors could interpret the findings more and generalize them into "theoretical implications" and "educational implications," especially for higher education supervisors or teachers and higher education students. In this way, the value and contribution of this research can be further demonstrated.
6.It is suggested that authors may state "Limitations and directions for future research".
Author Response
The authors appreciate and acknowledge the reviewer work and very meaningful comments. We respectfully answer to those questions :
The author improved comment 1 and did not sufficiently revise comments 2-6.
I recommend that the author indicate the line number of the revised text when answering each comment.
I hope the author's efforts will make this article better.
The authors thank and appreciate the comments and the possibility that was given to make this a better article, we also thank the reviewer for his contribution to achieving this goal.
- In the Introduction section: The author had organized a lot of literature, but the content of the literature is not systematic. I can’t understand the relationship between the research variables and the importance of this research.
This information is added on the lines 50 to 57, where is possible to read:
This impact is relevant to study and to look for relationship among the variables of literacy and mental health related to health promotion strategies, since literacy levels cam change according to study cycles, and it is recognized the relationship between higher levels of literacy and higher levels of creating mental health promotion strategies. This research is relevant to show if this happens in this specific population of alto Alentejo students. This may contribute to identify theoretical and educational strategies both for students and teachers in order to empower the higher education community for a better mental health
- Several issues of research method are as follows:
1)This article lacks a clear research model, research questions and research hypotheses.
The information on the research model is on the lines 182-184, where is possible to read:
This cross-sectional, descriptive, and correlational study used an online questionnaire administered to students from various study cycles at higher education institutions in Portugal.
The information on the research questions is on the lines 162-170, where is possible to read:
Considering, the above and the relationship among literacy, mental health and health promotion statergies, we have identified the following research questions:
(a) What is the association between mental health and health promotion strategies of higher education students in the Alentejo region of Portugal during confinement in the Covid19 pandemic?
- b) What is the correlation between self-reported academic success and mental health of higher education students in the Alentejo region of Portugal during confinement in the Covid19 pandemic?
- c) What is the association between the cycle of studies and mental health promotion strategy in higher education students in the Alentejo region of Portugal during confinement in the Covid19 pandemic?
The information on the research hypothesis is on the lines 185-187, where is possible to read:
The study hypothesis was to evaluate if there were any relationship among literacy and mental health of Portuguese higher education students and their use on health promotion strategies during confinement in the Covid19 pandemic.
2)The definition of independent and dependent variables was not clear. The author did not clearly define research variables such as "literacy", "health promotion strategies", "social network typology", "reasons for using social networks", "academic achievement", etc. The author did not explain how these variables are scored. Is it appropriate for students to self-assess their academic achievements? In addition, are the scores of these variables suitable for correlation analysis with mental health?
The information on these comments is on the lines 187-200, where is possible to read:
Independent variables were defined by literacy (ability to read, write, speak compute and solve problems at the level of proficiency, that allows the function in society and development of own’s knowledge and potential), health literacy (“the degree to which individuals can obtain, process, understand and communicate about health-related information needed to make informed health decisions”,[8]) and mental health (“a state of well-being in which the individual realizes his or her own abilities, can cope with the normal stresses of life, can work productively and fruitfully, and is able to make a contribution to his or her community”[13]) . Dependent variable is the use of health promotion strategies (processes to enable and to increase one’s control over the health in order to improve it) in the COVID 19 pandemic. Other independent variables are measure by the participation of the students in several activities groups of social network (network that facilitates social contact, leisure, and pleasure, in a huge variety of applications and activities, and academic achievement as the extent to which a student achieves educational goals..
3)The author did not specify the population and the specific sampling method.
The information on the population and specific sample method is on the lines 203-207, where is possible to read:
A non-probability convenience sampling was obtained, calculated for a 5% margin of error and a 90% confidence level, requiring 256 participants. The questionnaire was created using Google Forms and administered online via email with the access link for completion and sent to the 4450 students with active enrollment in the Institutions, and 329 valid questionnaires were obtained.
- The reliability and validity of research tools are very important. It is common to translate scale into a new language and use it in different countries. The author can explain the process of translating the original scale into Portuguese and how to ensure that items retain their original meaning after translation. It is common practice to use CFA (Confirmatory Factor Analysis) to further illustrate the validity of the scale.
The information on these comments is on the lines 217-224 and 235-241, where is possible to read:
The reduced version of the Mental Health Inventory (MHI-5), translated and validated for Portugal was used to assess Mental Health. Based on the thirty-eight question Inventory, a reduced version was developed called Mental Health Inventory 5 (MHI-5), composed of five items representing four dimensions of mental health: Anxiety, Depression, Loss of Emotional-Behavioral Control, and Psychological Well-Being (Ware et al., 1992). The scale's rating is obtained by summing the items (2 items with the rating reversed). Higher levels in the summation correspond to better mental health between 5 and 30.
These five items, have in the original study, a correlation of r=0.95 and r=0.92, with the total score of the 38-item version. The Portuguese adaptation shows a correlation of r=0.95 between the MHI-5 and the MHI-38. The scale score is obtained through the sum of the items (2 items with inverted scores). Higher levels in the sum correspond to better mental health between 5 and 30. Numerous investigations have shown that the MHI-5 is a useful screening test in the assessment of mental health [30].
- It is suggested that the authors could interpret the findings more and generalize them into "theoretical implications" and "educational implications," especially for higher education supervisors or teachers and higher education students. In this way, the value and contribution of this research can be further demonstrated.
The information on these comments is on the lines 398-408, where is possible to read:
The development of health literacy is crucial to the promotion of mental health, because better informed and better prepared students will surely be better able to face the adversities they will encounter. The universities must include mental health policies in their strategic plans. Specific diagnostic and intervention programs should be defined, as well as programs, to promote the mental health of university students. These intervention programs should be transversal to all students, with particular attention, to the most psychologically vulnerable students, female, younger, out-of-home, and with scholarships (usually the most financially needy) [47]. With this study we hope to contribute to the development of educational strategies, encouraging other researchers to further expand knowledge in this area of health strategies for mental health promotion.
6.It is suggested that authors may state "Limitations and directions for future research".
The information on these comments is on the lines 409-414, where is possible to read:
Besides been applied only to Portuguese students, another limitation to our study is the fact that it is a cross-sectional study, not allowing a cause-effect relationship. We cannot conclude from this study whether it is literacy and mental health that leads to adoption of health promotion strategies; however it shows the relevance to promote the mental health of higher education students to protect them and encourage a healthy lifestyle. Another limitation can be the impossibility to transpose these results to a situation other than a confinement for health reasons
Also on the lines 421-434, where is possible to read:
With this study, we hope to contribute to the development of educational strategies, encouraging other researchers to further expand knowledge in this area of health strategies for mental health promotion.
Besides been applied only to portuguese students, another limitation to our study is the fact that it is a cross-sectional study, not allowing a cause-effect relationship. We cannot conclude from this study whether it is literacy and mental health that leads to adoption of health promotion strategies; however, it shows the relevance to promote the mental health of higher education students to protect them and encourage a healthy lifestyle. Another limitation can be the impossibility to transpose these results to a situation other than a confinement for health reasons. Thus, evidence seems consistent that both higher years of study and field of study are related to mental health literacy, however, these studies are not common, and additional variables directly related to college experience should be investigated for a more comprehensive understanding of the factors associated with mental health literacy in the college population.
